# Mechanical Testing of Selective-Laser-Sintered Polyamide PA2200 Details: Analysis of Tensile Properties via Finite Element Method and Machine Learning Approaches

**DOI:** 10.3390/polym16060737

**Published:** 2024-03-08

**Authors:** Ivan Malashin, Dmitriy Martysyuk, Vadim Tynchenko, Vladimir Nelyub, Aleksei Borodulin, Andrey Galinovsky

**Affiliations:** Artificial Intelligence Technology Scientific and Education Center, Bauman Moscow State Technical University, 105005 Moscow, Russia; dmart9945@mail.ru (D.M.); vladimir.nelub@emtc.ru (V.N.); alexey.borodulin@emtc.ru (A.B.); a_galinovskiy@bmstu.ru (A.G.)

**Keywords:** polyamide, PA2200, SLS, tensile properties, additive manufacturing, polymer testing, finite element method

## Abstract

This study delves into the mechanical characteristics of polyamide PA2200 components crafted using selective laser sintering (SLS) technology. Our primary objective is to analyze the tensile behavior of the components printed at various orientations, showing its response to diverse loading conditions. Finite element method (FEM) modeling was employed to analyze the tensile behavior of these details. The time determined for breaking the detail is 9 s. In addition we forecast key properties, such as tensile behavior and strength, using machine learning (ML) techniques, and the best models are for predicting relative elongation are KNeighborsRegressor and SVR.

## 1. Introduction

Modern materials manufacturing technologies play a pivotal role in the development of innovative industrial solutions. Among them, the selective laser sintering (SLS) technology [1] stands out for its efficiency and versatile applications.

Modern innovations in materials manufacturing demand novel and efficient methods, with SLS emerging as a prominent representative. This additive manufacturing technique offers unique capabilities for creating intricate and durable structures from various materials. SLS technology is widely used in various industries due to its ability to create complex and functional parts. Below are some of the key application areas of SLS technology:Aviation and aerospace industry: SLS is used to manufacture critically important parts such as engine components, air and space structures, and for prototyping new designs. This helps to reduce design and development time, as well as to create components with optimal weight and strength characteristics [2].Medical industry: In the medical field, SLS is utilized for the production of customized implants, prosthetics, and anatomical models for surgical planning and training [3]. Its ability to create patient-specific solutions contributes to improved patient outcomes and medical advancements.Automotive industry: In automotive manufacturing, SLS is utilized for rapid prototyping of parts and functional components [4]. It facilitates the development of innovative designs and allows for the production of complex geometries with high precision, enhancing overall vehicle performance.Electronics: SLS technology is employed in the electronics industry for manufacturing housings, casings, and intricate components for electronic devices [5]. It enables the rapid production of prototypes and customized parts, contributing to the development of cutting-edge electronic products.Energy industry: Sectors involved in energy equipment production benefit from SLS technology for the fabrication of robust and high-precision components [6]. These components are essential for enhancing the efficiency and reliability of energy systems, including renewable energy technologies.Defense industry: SLS is instrumental in the defense sector for producing lightweight, durable, and complex parts for defense systems, including aerospace and ground-based applications [7]. It supports rapid prototyping, testing, and customization of components to meet specific defense requirements.Tool and equipment manufacturing: SLS technology is applied in tool and equipment manufacturing for the production of tooling [8], jigs, fixtures, and functional prototypes [9]. Its versatility and ability to create intricate geometries make it a valuable tool for various manufacturing applications.

At the core of SLS technology is the use of laser irradiation to sinter thin layers of powdered material, typically polymers [10]. This method allows precise layering and the creation of three-dimensional objects without the need for molding or the application of binding agents [11]. Consequently, SLS provides a high degree of flexibility in production, enabling the fabrication of highly complex components [12].

Key advantages of SLS technology include its high production speed, the absence of a need for supporting structures during object creation, and the ability to use a variety of materials, including polymers, metals, and composites [13].

Testing of SLS-produced details is a focal point in numerous scientific papers. For instance, Ref. [14] delves into the impact of dynamic tension/compression loading on SLS polyamide components. Through thermal and microstructural studies, it sheds light on fatigue phenomena, revealing crack initiation from inclusions caused by unfused powder particles and emphasizes the crucial role of material density in determining fatigue life, with a lower density correlating to increased crack initiation chances.

Ref. [15] explores the influence of four SLS parameters on the mechanical properties of printed parts, printed at varying angles (0°, 45°, and 90°). By varying the laser power, scanning speed, layer thickness, and scan spacing, we investigate their impact on tensile strength, modulus of elasticity, and elongation at break. Tensile tests on specimens printed at different angles provide insights into strength and stiffness, while regression models offer pathways for enhancing future SLS parts.

The orientation of parts during printing, alongside various manufacturing parameters, significantly influences the mechanical properties of the final products. Ref. [16] explores the influence of horizontal (H) and vertical (V) printing orientations on the ultimate tensile strength (UTS) of PA12 specimens. Our findings reveal that H specimens display greater deformations and reduced UTS scatter compared to V specimens, with the latter exhibiting a higher elastic modulus.

Ref. [17] investigates whether varying processing parameters at different heights along the z-axis of the working chamber maintain consistent mechanical properties and dimensional stability in 3D-printed parts, crucial for optimizing printing costs.

The impact of SLS parameters, including laser power, scan speed, hatch spacing, and scan length, on the mechanical properties of polyamide-12-printed parts is highlighted in [18]. The results suggest that hatch spacing is the most influential variable, followed by laser power and scan speed, in determining strength, modulus of elasticity, and elongation. Achieving optimal mechanical behavior while ensuring economical production poses a challenge, as higher strength parts require lower hatch spacing and slower scan speeds. Additionally, excessive laser energy density can lead to powder burning and printing failure. The scan length minimally affects the modulus of elasticity and insignificantly impacts strength and elongation.

Ref. [19] addresses this gap by examining the influence of desktop-SLS-printed parts’ orientations and diameters on their structural and mechanical parameters. The results indicate that for parts subjected to tensile load optimal mechanical parameters are achieved when printed at a 0° angle, while the diameter of printed elements significantly influences their geometric and dimensional representation.

An investigation on the effects of energy density and layer thickness on the mechanical properties, roughness, density, and particle melt degree of SLS PA12 tensile bars is described in [20]. The findings reveal an interaction between energy density and layer count, impacting fracture behavior and mechanical properties, thereby aiding in predicting optimal part thickness for thin-part dimensioning in laser-sintering applications.

Through a design of experiments approach, regression models were developed in [21] to elucidate the relationship between process settings and part properties, revealing significant impacts of layer thickness and scan spacing variations on mechanical properties.

Similarly to PA12, the primary application of PA2200 lies in additive manufacturing or 3D printing [22]. The material’s favorable properties, such as its ability to be processed into fine powders and its excellent layer adhesion during printing, contribute to its success in this domain. Additive manufacturing processes utilizing PA2200 enable the production of components with enhanced mechanical performance.

Furthermore, PA2200 finds extensive use in the automotive industry, particularly in the manufacturing of lightweight and durable components. Its low moisture absorption, coupled with excellent dimensional stability [23], makes it an ideal choice for applications where exposure to varying environmental conditions is inevitable. The material’s contribution to reductions in vehicle weight, leading to improved fuel efficiency, underscores its significance in the pursuit of sustainable and eco-friendly transportation solutions [24].

It is also important to note several distinctive properties of PA2200:PA2200 demonstrates high resistance to oils, fats, various solvents, and other chemical substances, ensuring stability and reliability in different environments [25].PA2200 exhibits significant thermal resistance, maintaining stability even at high temperatures [26]. This quality makes it relevant for applications involving elevated temperatures and processes where the material is exposed to high-intensity energy radiation.PA2200 combines flexibility and strength, allowing it to withstand deformation and high impact loads without compromising its structural integrity [27].

Forecasting the mechanical attributes of materials is integral for enhancing material efficacy across various applications. As industries increasingly turn to advanced manufacturing techniques, the ability to forecast key properties via ML becomes paramount. ML has emerged as a powerful tool for predicting and understanding material behavior [28]. We delve into the importance of predictive modeling in the context of materials engineering, shedding light on how machine learning methodologies contribute to a deeper comprehension of material properties and provide essential insights for optimizing manufacturing processes.

Our study focuses on analyzing the mechanical properties of components produced from polyamide PA2200, a refined variant of polyamide 12 (PA 12) [29], also known as Nylon 12, through SLS. Derived from the polymerization of ω-aminolauric acid or its lactam monomer [30], it exhibits a high melting point, making it well suited for applications demanding thermal endurance [31]. We compare the tensile properties at various printing angles, including printing on edges. Additionally, tensile tests are conducted, followed by modeling the results using FEM. Also we propose an ML approach that can predict characteristics based on available data.

## 2. Materials and Methods

### 2.1. Fabrication Process

Three-dimensional models of future products were created using CAD program Autodesk Inventor 2023. The sample’s length, denoted by B, was consistently 10 cm, while the thickness, indicated by H, ranged from 0.61 to 0.71 cm during printing. Notably, these dimensions were obtained for samples printed not on an edge. Furthermore, the average cross-sectional area, $A_{0}$, of the samples was approximately 6.4 cm^2^. Conversely, for details printed on an edge, the average thickness of the component was 0.57 cm, with an average length of 10.25 cm and a cross-sectional area, $A_{0}$, of 5.84 cm^2^. All dimensional data regarding the specimens area contained in Table 1 and Table 2.

Figure 1 shows a virtual representation of intricately designed polyamide components fabricated through the additive manufacturing technique of selective laser sintering (SLS).

Upon the conclusion of the modeling phase, the process proceeded to the configuration of the printing parameters. Insights gained from prior research and experimentation aided in discerning the optimal settings for 3D printing of PA2200 polyamide. These parameters included an extruder temperature of 190 °C, printing speed of up to 70 $\frac{\mathrm{mm}}{s}$, and a density of 0.85 g/cm^3^. The average particle size was 60 microns, with a particle size range of 20–80 microns. The bulk density at 20 °C was 0.70 g/cm^3^, and the dehydration level at 23 °C was 0.05%. The melting temperature, measured by differential scanning calorimetry (DSC), was 182 °C, with a breaking point or failure point of 220 °C. The hardness on the Shore scale was 88 units. The tensile strength was 45 MPa, with a fracture strain of 5%. The ash content was 0.1%, and the tensile modulus was 1700 MPa.

Ensuring the proper functioning of the 3D printer, preparations for printing commenced. The loading of the PA2200 material was executed with meticulous attention to minimize potential defects on the surface of the prospective specimen. Subsequently, adjustments to the bed and extruder were made, with the optimization of parameters for the first layer and bed level for sustainable formation of each layer and the establishment of a robust foundation for subsequent layers.

With the sequence of adjustments concluded, the actual 3D printing process was initiated. The commencement was orchestrated in consideration of the recommended parameters, and vigilant monitoring of the process’s onset is imperative for identifying potential issues that might impact the results of the subsequent analysis.

Following the completion of the printing, post-processing steps were undertaken. This stage involved the meticulous removal of excess material and support structures. Additional procedures such as grinding and polishing may be applied based on the desired surface finish.

It is crucial to underscore that each of the described stages is intricately interconnected, and only the scrupulous execution of all steps ensures the creation of a high-quality specimen for subsequent analysis. An engineering approach, amalgamating 3D printing and analysis within software environments, facilitates the attainment of optimal results in the development and testing of materials and structures.

### 2.2. Finite Element Method Modeling

The finite element method (FEM) [32], a numerical technique in the field of engineering, serves as a computational tool for the meticulous examination and resolution of deformable solid mechanics problems. When applied to polymers like polyamide PA2200, FEM facilitates detailed modeling and calculation of the material’s physico-mechanical characteristics. This method involves decomposing complex structures into finite elements, followed by the numerical solution of equations governing deformable solid mechanics [33].

In this section, we present the finite element method (FEM) modeling conducted for the analysis of a polyamide 2200 component. The simulation, conducted using COMSOL Multiphysics 6.2, utilized mesh parameters with 4272 elements, including 24 vertex elements, 270 edge elements, and 1816 boundary elements. The minimum element quality was set to 0.3041. Tensile testing was performed, and stress–strain curves were generated to analyze the material’s mechanical response. The resulting stress distribution diagrams are shown in Figure 2. They offer insights into localized stress concentrations, aiding in the assessment of structural integrity and deformation patterns.

The FEM model was specifically designed to study the mechanical behavior of the polyamide 2200 component under tensile loading conditions. The chosen mesh parameters ensure a sufficient level of detail in capturing the geometry and intricacies of the component.

A tensile test was conducted to evaluate the response of the polyamide 2200 component to applied tensile forces. The model was subjected to controlled tensile loading, and stress–strain diagrams were generated to analyze the material’s mechanical properties under various conditions.

Stress distribution diagrams were obtained to visualize the variations in stress across the polyamide 2200 component during the tensile test. These diagrams provide insights into regions experiencing high or low stress concentrations, aiding in the identification of potential failure points and deformation patterns.

The results of the FEM simulation, including stress–strain curves and stress distribution diagrams, contribute to understanding of the mechanical response of the polyamide 2200 component. These findings are essential for optimizing the design and performance of components manufactured from polyamide 2200.

### 2.3. Experimental Methodology: Conducting Tensile Tests on PA2200 Material Using the QUASAR 50 Universal Testing Machine

We employed the QUASAR 50 universal testing machine (Galdabini, Cardano al Campo, Italy) to perform tensile tests [34] on specimens made of the PA2200 material. The initial phase involved preparation of the specimens to ensure compliance with established standards or modification in accordance with unique research requirements. An integral aspect of this phase was processing of the specimens, with particular attention to their dimensions and form.

The subsequent procedure involved the meticulous mounting of the specimens onto the QUASAR 50 testing apparatus. The selection of this equipment was driven by its superior efficiency and precision in delivering accurate outcomes. A critical aspect of this phase was ensuring the secure fixation of the specimens and precise alignment within the machine. Subsequently, various tests were configured. Diagrams illustrating the relationship between applied force and specimen elongation (traverse position) were obtained. Figure 3 depicts a representative example of such a diagram for specimens made of PA2200 polyamide, printed at a 0° orientation.

Initiation of the testing process involved activating the machine in accordance with pre-established parameters. Throughout the course of the QUASAR 50 test, systematic recording of force and deformation changes took place, while the researcher vigilantly monitored the evolution of specimen characteristics. This encompassed tracking elasticity, yield strength, and fracture moment.

Upon completion of the test, an analysis of the obtained data was conducted to compute values for strength, elasticity, and other key characteristics of the PA2200 material.

It is important to note that specific procedural steps may vary slightly depending on the configuration of the utilized testing machine and the requirements of the specific test. Strict adherence to equipment operating instructions and the application of standard testing methods were maintained throughout the testing process to ensure high reliability and precision of the obtained results.

## 3. Results

The tensile loading test was conducted using the QUASAR 50 universal testing machine, and force–displacement diagrams were obtained. Figure 4 illustrates an example of such a diagram for PA2200 polyamide samples printed (a) with $0^{\circ }$, $45^{\circ }$, and $90^{\circ }$ orientations and (b) with $0^{\circ }$ and $90^{\circ }$ edge orientations. Curves labeled as ’approximated’ represent results obtained through FEM simulations.

From the diagrams, the following characteristics were derived: $F_{\mathrm{pl}}$—the force at the proportional limit, $F_{0.2}$—the force at the yield point (where the sample elongation reaches 0.2% of the original length $l_{0}$), $F_{\max}$—the force at sample failure, and $\Delta l_{\max}$—the maximum sample elongation at failure.

Subsequently, the following calculations were performed:$$\sigma_{\mathrm{pl}}=\frac{F_{\mathrm{pl}}}{A_{0}}$$
where $\sigma_{\mathrm{pc}}$ is the proportional limit, and $A_{0}$ is the cross-sectional area of the sample.
$$\sigma_{0.2}=\frac{F_{0.2}}{A_{0}}\;$$
where $\sigma_{0.2}$ is the yield strength.
$$\sigma_{\max}=\frac{F_{\max}}{A_{0}}$$
where $\sigma_{\max}$ is the ultimate tensile strength.
$$\varepsilon =\frac{\Delta l_{\max}}{l_{0}}$$
where ε is the relative elongation at failure.

The calculation results for PA2200 polyamide samples in the Oxy plane are presented in Table 1 and Table 2, where Ψ represents the orientation angle of the samples in the Oxy and Ozy planes, respectively, during printing.

The data presented in these tables provide intriguing insights into the mechanical properties of SLS-produced details. Specifically, multiple details (3 or 4 for each experiment) were printed at various angles and orientations, and their mechanical characteristics were subsequently measured and computed. These variations in printing parameters can significantly influence the structural integrity and performance of the final product.

For instance, upon examining the resulting characteristics, including tensile strength ($F_{ts}$), the force at the yield point ($F_{0.2}$), and maximum load ($F_{max}$), noticeable trends emerge based on the printing orientation. Notably, details printed on their edges exhibit lower values for these characteristics compared to ordinary orientation.

Additionally, the printing orientation can significantly influence other properties, such as the elongation at break, denoted by ε. Interestingly, this parameter tends to be approximately 6% lower when details are printed on their edges compared to the typically observed values for other orientations.

By elucidating the correlations between printing parameters and resulting mechanical characteristics, our study contributes to the advancement of additive manufacturing techniques and underscores the importance of tailored approaches for achieving desired material properties. Such insights pave the way for informed decision making in material design and process optimization, ultimately enhancing the reliability and functionality of SLS-produced components in real-world applications.

### ML Approach for Predicting Calculated Characteristics

In this study, machine learning (ML) regression models were employed to predict various characteristics of PA2200 specimens using different regression algorithms. The application of ML facilitated accurate predictions of critical mechanical properties, including tensile strength ($F_{ts}$, N), yield point stress ($\sigma_{pl}$, MPa), yield point force ($F_{0.2}$, N), yield point stress ($\sigma_{0.2}$, MPa), maximum load ($F_{max}$, N), tensile stress ($\sigma_{ts}$, MPa), and elongation at break (ε, %). By leveraging algorithms like linear regression, decision tree regressor, random forest regressor, and others, we were able to capture complex relationships between process parameters and material properties.

We systematically applied 10 regressors to predict each of the mentioned seven characteristics, resulting in a total of 70 experiments (each cell in Table 3 represents a result for each experiment). For instance, we predicted elongation at break (ε, %) using one of the regressors, given the remaining characteristics. This process was repeated for all characteristics, with each cell in the table representing the prediction result on the dataset from Table 1. We adopted a train/test split ratio [35] of 80/20% to ensure robust model training and evaluation.

These predictions provide information into the material’s performance under varying conditions and loading scenarios, aiding in the selection of the most suitable regressor for a particular characteristic. Table 3 presents a summary of all of the employed regression models for each calculated characteristic. A lower score indicates a better prediction of the regression model.

As evident from the table, the KNeighborsRegressor and SVR models exhibit the most consistent performance, both achieving an MSE score of 0.12 in predicting relative elongation.

The appropriate selection of a regressor will enable researchers to determine the desired characteristic value of a component printed at a specific angle with defined dimensions. By inputting all necessary characteristics into the model, researchers can predict the value of the parameter of interest, thus facilitating informed decision making in additive manufacturing processes.

## 4. Discussion

The mechanical properties of PA2200 polyamide details fabricated via SLS were systematically investigated in this study to gain insights into their behavior under various loading conditions. The focus was on understanding how different printing orientations (0°, 45°, and 90°) influence the tensile behavior and strength of the fabricated specimens.

Referring to the data presented in Table 1, it is evident that the elongation at break (ε) and ultimate tensile strength ($\sigma_{ts}$) are higher when printed at a 45° angle. However, there is a noticeable trend across increasing printing orientations where the maximum values of the tensile strength ($F_{ts}$), yield point force ($F_{0.2}$), and maximum load ($F_{max}$) show an upward trend. Specifically, samples printed at certain angles exhibit higher or lower values for these parameters compared to others, indicating the influence of printing orientation on the mechanical performance of the parts.

Printing parts on the edge, as our research has shown, is accompanied by a significant decrease in the average values of $F_{ts}$, ranging from 22 to 28 N. This result is considerably lower than when printing in the standard orientation, where $F_{ts}$ values are approximately 40–48 N. One potential reason is the altered cooling rate during the printing process [45], leading to differences in material crystallinity and interlayer bonding. Additionally, the orientation of the layers may affect the distribution of internal stresses [46] within the part, potentially weakening its overall mechanical integrity. Further investigation into the microstructural changes and printing parameters is necessary to fully understand the underlying mechanisms [47].

It is essential to acknowledge the potential influence of various methodological factors on the obtained results. Factors such as variations in printing parameters, layer thickness, and post-processing techniques may have contributed to the observed differences in mechanical properties.

The insights gained from this study have important implications for the design and optimization of SLS manufacturing processes. By understanding how printing orientation affects mechanical properties, manufacturers can tailor their printing strategies to achieve desired performance characteristics. Future research endeavors could focus on further elucidating the underlying mechanisms governing the relationship between printing orientation and mechanical behavior, as well as exploring advanced computational modeling techniques to predict material properties with greater accuracy.

During our investigation into the mechanical properties of SLS-manufactured products, particularly those made from polyamide PA2200, we observed a notable influence of the design software package used, COMSOL. This program, however, demonstrated inherent limitations, hindering its full potential. A major constraint lies in its restricted ability to modify material properties, especially for polymers like polyamides. This limitation complicates precise modeling in practical engineering situations, as adjustments to chemical and physical properties are constrained within the available options [48].

Specific limitations associated with integrating results from natural experiments into software packages create obstacles for a more accurate analysis corresponding to real conditions. Difficulties arise in integrating data obtained from experiments on physical samples into the software [49].

Taking into account the identified limitations, recommendations and optimization strategies are presented to expand the analytical space, especially in the field of materials, including polymers. The development of more flexible tools for editing material properties and the improvement of mechanisms for integrating experimental data, considering diverse data formats and standards, could significantly enhance the accuracy and applicability of software packages for various engineering tasks.

This research emphasizes the imperative for continuous improvement in design software platforms, providing engineers with more refined tools for mechanical analyses and optimization of polymer and other material components. The proposed recommendations advocate for the integration of empirical data, such as polymer characteristics gleaned post-theoretical design, into these software platforms. The incorporation of specific data and chemical formulations within material properties represents a notable stride in refining the analysis of physico-mechanical properties of polymers. Furthermore, a progressive approach involves the enhancement of finite element structures and the augmentation of the toolkit for manipulating finite element geometry. Despite potential computational overhead, this approach substantially mitigates the risk of errors in digital product configuration and modeling, thereby elevating the precision of subsequent calculations and simulations.

## 5. Conclusions

The investigation into the mechanical properties of polyamide PA2200 components manufactured via selective laser sintering has revealed insights into the interplay between printing orientations, material characteristics, and structural integrity.

Our experiments revealed notable variations in the mechanical properties of the tested elements depending on the printing orientation. Specifically, we observed that specimens printed at different angles exhibited distinct tensile strength, elongation at break, and other mechanical characteristics. For components printed on an edge, the average elongation at break was observed to be 6% lower compared to those printed in the standard orientation. This underscores the importance of considering the print direction in the design and manufacturing of components via selective laser sintering using polyamide PA2200.

The experimental results were instrumental in validating the accuracy of FEM simulations conducted using the COMSOL Multiphysics software. By comparing the experimental data with the simulated outcomes, we confirmed the reliability of our numerical models in predicting the mechanical behavior of SLS-produced components. In addition, ML methods were harnessed to identify the optimal regressor for forecasting the specific attribute of a component printed at a particular angle and with predetermined dimensions. This approach allows researchers to input all relevant characteristics into the model, enabling the anticipation of the parameter’s value of interest.

In summary, the incorporation of ML and FEM methodologies constitutes an indispensable step in improving engineering design processes, enhancing the results’ accuracy, overcoming limitations of software packages, and actively fostering the development of innovative approaches in the mechanical analysis of products made from polymers. 

## Figures and Tables

**Figure 1 polymers-16-00737-f001:**
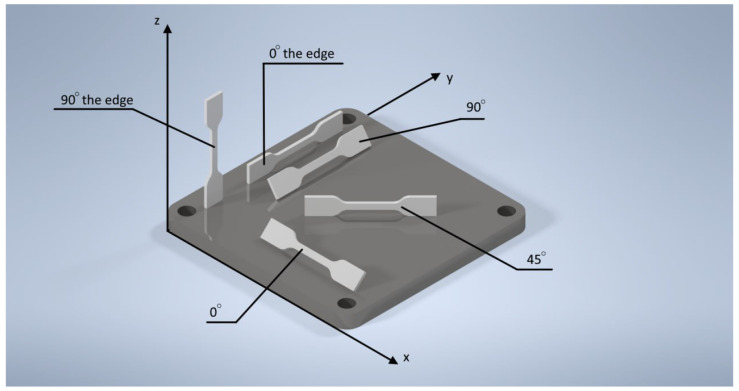
Three-dimensional models of polyamide components produced via selective laser sintering (SLS).

**Figure 2 polymers-16-00737-f002:**
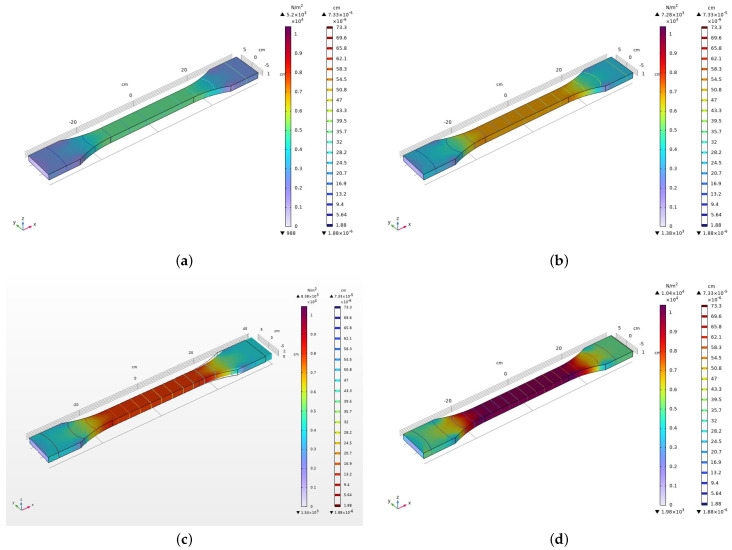
FEM modeling outcomes depicting stress distribution and deformation characteristics of the polyamide 2200 component under tensile loading conditions. The detailed analysis captures the component’s behavior at specific time instances: (**a**) 0 s, (**b**) 3 s, (**c**) 6 s, (**d**) 8 s.

**Figure 3 polymers-16-00737-f003:**
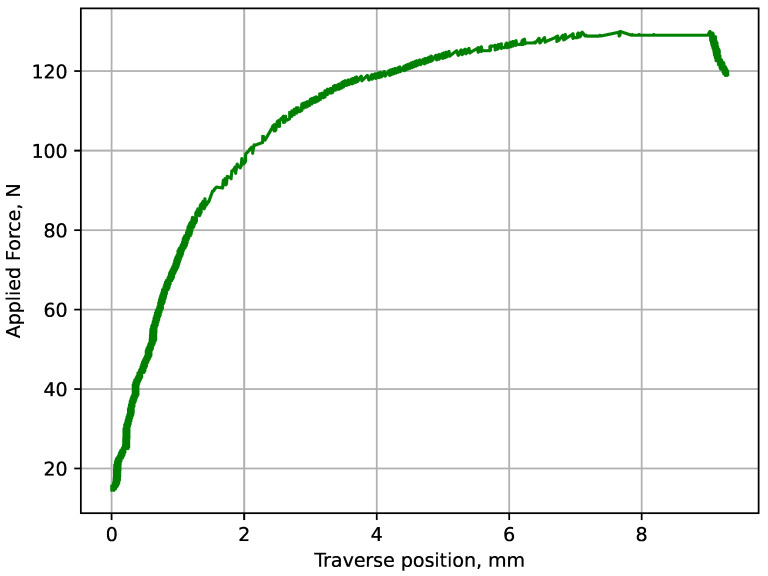
The dependence of the applied force on the position of the crosshead for a sample made of PA2200 polyamide, printed at 0°.

**Figure 4 polymers-16-00737-f004:**
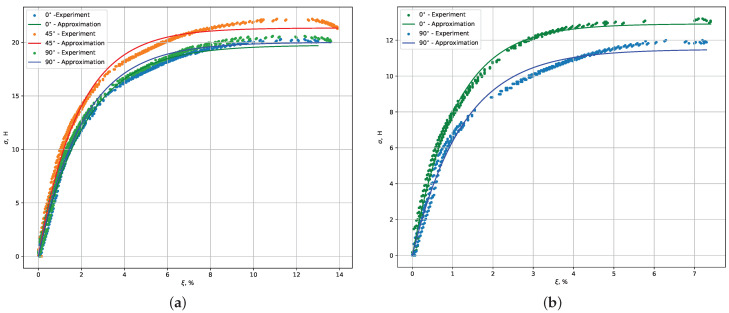
Dependency of the load force on the position of the traverse for a sample made of The following highlights are the same. PA2200 polyamide, printed at (**a**) $0^{\circ }$, $45^{\circ }$, and $90^{\circ }$ orientations and (**b**) $0^{\circ }$ and $90^{\circ }$ edge orientations. ξ represents the relative elongation, and σ represents the applied force. The points represent experimental data obtained from the tensile testing setup, while the lines represent the approximation generated by FEM model developed in COMSOL.

**Table 1 polymers-16-00737-t001:** Results for samples made of PA2200 polyamide with printing orientations at $0^{\circ }$, $45^{\circ }$, and $90^{\circ }$.

Ψ,°	№	H, cm	B, cm	$A_{0}$, ${\mathrm{cm}}^{2}$	$F_{\mathrm{ts}},N$	$\sigma_{\mathrm{pl}}$, MPa	$F_{0.2}$, N	$\sigma_{0.2}$, MPa	$F_{\max}$, N	$\sigma_{\mathrm{ts}}$, MPa	ε,%
0	1	0.64	10.1	6.46	44	6.8	68	10.5	131	20.3	14
	2	0.63	10	6.3	46	7.3	67.3	10.7	128	20.4	13.6
	3	0.63	10	6.3	44.5	7.06	61.3	9.7	127.5	20.3	14
45	1	0.64	10	6.4	46.6	7.28	72.3	11.3	142.6	22.3	14.6
	2	0.61	10	6.1	45.7	7.5	73.8	12.01	137.6	22.6	14.6
	3	0.63	10	6.3	50	7.93	77.7	12.3	142	22.6	13.4
	4	0.63	10	6.3	46.6	7.4	62.7	9.95	134.8	21.2	12.8
90	1	0.71	10	7.1	48.7	6.86	73.3	10.32	143.2	20.17	13.4
	2	0.64	10	6.4	45	7.03	68.4	10.69	129.4	20.22	13.4
	3	0.62	10	6.2	49.4	7.97	68.4	11.03	135.7	21.89	14
	4	0.64	10	6.4	47.2	7.38	69	10.78	129.6	20.25	13.7

**Table 2 polymers-16-00737-t002:** Results for samples with printing orientations at $0^{\circ }$ and $90^{\circ }$ to edge.

Ψ,°	№	H, cm	B, cm	$A_{0}$, ${\mathrm{cm}}^{2}$	$F_{\mathrm{ts}},N$	$\sigma_{\mathrm{pl}}$, MPa	$F_{0.2}$, N	$\sigma_{0.2}$, MPa	$F_{\max}$, N	$\sigma_{\mathrm{ts}}$, MPa	ε,%
0	1	0.57	10.25	5.84	28	4.79	42.8	7.33	78.4	13.42	8.8
	2	0.57	10.3	5.87	30.5	5.20	43.4	7.39	73.3	12.49	7.1
	3	0.57	10.25	5.84	26.4	4.52	43.1	7.38	73.6	12.60	5.9
	4	0.57	10.25	5.84	33.1	5.67	47.8	8.18	84.5	14.46	7.7
90	1	0.56	10.25	5.74	27.5	4.79	38.9	6.78	72.4	12.61	7.3
	2	0.53	10.25	5.43	23.2	4.27	34.4	6.33	61.6	11.34	7.3
	3	0.53	10.25	5.43	22.9	4.22	36.5	6.72	65.7	12.09	7.4

**Table 3 polymers-16-00737-t003:** Regression results—MSE scores [36] for calculated characteristics.

Regressor	$F_{\mathrm{ts}},N$	$\sigma_{\mathrm{pl}}$, MPa	$F_{0.2}$, N	$\sigma_{0.2}$, MPa	$F_{\max}$, N	$\sigma_{\mathrm{ts}}$, MPa	ε,%
LinearRegression [29]	0.10	0.00	0.23	0.01	0.93	0.02	2.38
DecisionTreeRegressor [37]	18.09	0.24	13.41	1.35	15.52	0.16	1.08
RandomForestRegressor [38]	9.06	0.28	11.93	0.58	9.84	0.62	0.31
GradientBoostingRegressor [39]	10.26	0.28	15.21	0.53	7.31	0.94	0.52
SVR [40]	7.69	0.38	29.94	0.90	35.35	2.24	0.12
KNeighborsRegressor [41]	6.80	0.41	19.10	0.72	9.79	1.27	0.12
MLPRegressor [42]	69.33	0.62	86.87	2.26	12.54	90.99	3.01
Ridge [43]	6.32	0.04	8.36	0.03	21.84	0.71	0.63
Lasso [44]	5.99	0.43	10.83	0.35	20.17	1.03	0.23

## Data Availability

Data are contained within the article.

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
