# Peer review of "Mechanical Testing of Selective-Laser-Sintered Polyamide PA2200 Details: Analysis of Tensile Properties via Finite Element Method and Machine Learning Approaches"

_polymers, 2024, doi:10.3390/polym16060737_

Round 1

Reviewer 1 Report

Comments and Suggestions for Authors

This manuscript reports on the mechanical properties of polyamide 12 (PA 2200) specimens processed by additive manufacturing (AM) using selective laser sintering (SLS). The authors analyzed the material’s performance under different loading conditions to obtain interesting data that are shown in Appendix A.

The tensile behavior of the specimens was then simulated by finite element method (FEM) to forecast the key properties as tensile behavior and strength by using machine learning (ML) techniques. The authors considered that the most sustainable models are KNeighborsRegressor and SVR for predicting relative elongation. However, no reasoning is given in the text for leading to this conclusion. In addition, no comparison between the simulated data and the experimental data is shown. The whole content of the manuscript seems to be composed of conceptual and methodological descriptions without an explanation of the data and related discussion. The authors are strongly requested to improve the discussion by explaining the data in detail. Otherwise, this manuscript may not be treated as a scientific paper.

The following points should also be reconsidered.

1)    The use of abbreviated names in the title and abstract are unfavorable. Full names should be added to them.

2)    L. 131-133, “Polyamide PA 2200 originates from the polymerization of caprolactam (Fig. 3b), a cyclic ester derived from caproic acid. The polymerization process of caprolactam results in the formation of extensive polyamide chains, constituting the high-molecular-weight structure of polymer PA 2200.” : This explanation including Figure 3 is wrong. “caprolactam” should be ω-aminolauric acid or its lactam monomer.

3)    The data shown in Appendix A are very interesting and must be explained in the text.

Author Response

Dear Reviewer1,

I hope this message finds you well. I am writing to express my sincere gratitude for the thorough and insightful review you provided for our manuscript.

Your expertise and attention to detail greatly contributed to the refinement of our work. Your constructive feedback and valuable suggestions have been instrumental in enhancing the quality and clarity of the manuscript. We truly appreciate the time and effort you dedicated to evaluating our research.

We have carefully considered each of your comments and have made the necessary revisions to address them effectively. 

Once again, thank you for your invaluable contribution to our research. We look forward to the opportunity to submit the revised manuscript for your further consideration.

Warm regards,

Ivan Malashin.

Reviewer 2 Report

Comments and Suggestions for Authors

Dear Authors,

The properties of polyamide materials produced by SLS technology are well-known and widely described in many scientific articles. In this respect, the article presented for review brings nothing new. Moreover, the article contains many general, non-specific formulations not used in technical scientific articles. The entire chapter 1 is more suitable for a textbook than for a technical research article. It is unknown why chapter 2.2 was included, which contains rather encyclopedic information. The cited literature is incomplete. In recent years, many publications have been published on research on polyamide materials produced by additive technology. Below are some detailed comments:

1. The abbreviation "FEM&ML" should not be used in the title of the article. Generally, titles should not use abbreviations or formulas. What does the abbreviation ML also mean in the Abstract and further parts of the article?

2. E-numbering 1-7, Lines 22-36 is a typical textbook text.

3. Line 81 – no literature citation.

4. Chapter 2 - the dimensions of the samples and the standard based on which the samples were made were not provided.

5. Chapter 2 - the technology of specimens preparation is not described. What printer was used to make the samples? What powder was used? What technological parameters were used, e.g. laser power, layer thickness, etc? This type of data is standardly provided in scientific articles.

6. Chapter 2.3 - no experimental parameters are given, e.g. the applied strain rate. The numbers of the standards used were not provided. It was only stated that appropriate standards were applied. This method of description is unacceptable in scientific articles.

7. The caption under Figure 2 should be short. Characteristics of the simulation should be included in the text.

8. Line 151 – what does the question mark mean in cited literature?

9. Figure 4 is unnecessary and adds nothing to the content of the article.

Summary: The article in its current form is not suitable for publication.

Kind regards

Reviewer

Author Response

Dear Reviewer 2,

I hope this message finds you well. I wanted to take a moment to express my deepest appreciation for the thorough and insightful review you provided for our manuscript titled.

Your meticulous attention to detail and constructive feedback have been incredibly valuable to us. Your expertise in the field has helped us identify areas for improvement and refine the clarity and rigor of our research.

We have carefully considered each of your suggestions and have made revisions accordingly to enhance the quality of our manuscript. 

Thank you once again for your time, expertise, and professionalism. Your efforts have undoubtedly strengthened our manuscript, and we sincerely appreciate your contributions to our work.

Warmest regards,

[Your Name]

Round 2

Reviewer 1 Report

Comments and Suggestions for Authors

The authors have revised their manuscript by considering the reviewer's comments. The manuscript can be accepted for publication.

Author Response

Dear Reviewer 1,

Thank you for your prompt response. We are pleased to hear that the revisions have met your expectations. We appreciate your thorough review of our manuscript and are grateful for your positive evaluation. We look forward to seeing our work published and contributing to the advancement of our field. 

Best regards, Ivan Malashin.

Reviewer 2 Report

Comments and Suggestions for Authors

Dear Authors,

            Thank you for making corrections according to my suggestions. The new, better text of the summary allowed me to make one more observation regarding the conclusions.

            The abstract states "Our primary objective is to analyze the tensile behavior of the components printed at various orientations, showing its response to diverse loading conditions", while in Chapter 5. Conclusion nothing is written about the results of the analysis of the influence of the orientation of printed elements on the mechanical properties. This means that the intended purpose of the work was not achieved. Chapter 5 describes in general terms the importance of FEM and ML methods. However, these are not conclusions. Conclusions should be specific and based on conducted experiments. I propose writing new conclusions based on the research completed. Chapter 5 should include answers to the following questions: what is the impact of print directions on the mechanical properties of the tested elements? How were the experimental results related to FEM and ML methods?

Kind Regards

Reviewer

Author Response

Dear Reviewer,

Thank you for your valuable feedback and for highlighting the discrepancy between the stated objective in the abstract and the content of Chapter 5. We acknowledge the importance of aligning the conclusions with the objectives and findings of the study. We have revised Chapter 5 to include specific conclusions based on the experimental results regarding the impact of print directions on the mechanical properties of the tested elements. Additionally, we provided a detailed discussion on how the experimental results were related to the Finite Element Method (FEM) and Machine Learning (ML) methods. Your insights were greatly appreciated, and we were committed to ensuring that the conclusions accurately reflected the outcomes of our research. (lines 360-381)

Thank you once again for your time and dedication to reviewing our manuscript. We look forward to incorporating your feedback and submitting the revised version for your further evaluation.

Best regards, Ivan Malashin.
